# ADVANCING ALGORITHMIC TRADING WITH LARGE LANGUAGE MODELS: A DEEP REINFORCEMENT LEARNING APPROACH FOR STOCK MARKET OPTIMIZATION

## ABSTRACT

In the fast-evolving landscape of financial markets, effective decision-making tools are essential for managing complexities driven by economic indicators and market dynamics. Algorithmic trading strategies have gained prominence for their ability to execute trades autonomously, with Deep Reinforcement Learning (DRL) emerging as a key approach for optimizing trading actions through continuous market interaction. However, RL-based systems face significant challenges, particularly in adapting to evolving time series data and incorporating unstructured textual information. In response to these limitations, recent advancements in Large Language Models (LLMs) offer new opportunities. LLMs possess the capacity to analyze vast volumes of data, providing enhanced insights that can complement traditional market analysis. This study proposes a novel approach that integrates six distinct LLMs into algorithmic trading frameworks, developing Stock-Evol-Instruct, an innovative instruction generation algorithm. This algorithm enables RL agents to fine-tune their trading strategies by leveraging LLM-driven insights for daily stock trading decisions. Empirical evaluation using real-world stock data from Silver and JPMorgan demonstrates the significant potential of this approach to outperform conventional trading models. By bridging the gap between LLMs and RL in algorithmic trading, this study contributes to a new frontier in financial technology, setting the stage for future advancements in autonomous trading systems.

## 1 INTRODUCTION

Financial markets, shaped by a complex interplay of factors such as economic indicators and investor behavior, require sophisticated decision-making tools to navigate their inherent volatility (Ashtiani & Raahemi, 2023a). Algorithmic trading strategies, which automate the execution of stock trading decisions, have emerged as crucial mechanisms in modern financial markets (Gurung et al., 2024). These strategies, driven by advanced computational algorithms, operate autonomously and have gained significant attention from investors and financial analysts complexity of influencing stock prices. Advances in information technology and machine learning have further revolutionized algorithmic trading, enabling more precise and timely decisions without direct supervision (Treleaven et al., 2013). However, the potential risks associated with these systems remain profound, as poorly algorithms can lead to catastrophic financial outcomes (Tudor & Sova, 2024). A critical challenge lies in the ability of these algorithms to dynamically adapt to the continuously evolving price time series, necessitating the development of more intelligent, flexible decision-making frameworks to maintain efficacy in an ever-changing market environment (Théate & Ernst, 2021).

Conventional algorithmic trading strategies, such as trend-following and mean reversion, have long provided the structural backbone for modern trading methodologies (Tudor & Sova, 2024). However, the advent of machine learning—specifically supervised learning and reinforcement learning (RL)—has revolutionized the field. Supervised learning models excel in forecasting stock trends through the analysis of structured historical data, while RL optimizes trading decisions by continuously interacting with the market, refining strategies through iterative feedback loops (Lei et al.,

2020; Sutton, 2018). The adaptive nature of RL is particularly well-suited to complex environments, such as those involving intricate pattern recognition (Alkhamees & Aloud, 2021), and has proven pivotal in advancing algorithmic trading systems (Bertoluzzo & Corazza, 2012). Despite its potential, RL faces significant challenges, including the management of irregularly spaced price time-series data (Glattfelder et al., 2011; Weerakody et al., 2021), efficient feature selection from an expansive search space (Moody & Saffell, 2001), and the inherent complexity of machine learning models (Lei et al., 2020; Kumar et al., 2021). In response to these challenges, the Directional Change (DC) event-based approach, which leverages intrinsic time to more accurately capture market dynamics, offers a promising alternative to traditional methods. However, its application remains constrained by its inability to adapt effectively to both small and extremely large datasets, limiting its scalability and broader utility in diverse tradings (Alkhamees & Aloud, 2021; Tsang et al., 2024).

Simultaneously, large language models (LLMs), such as OpenAI's GPT-based (Brown, 2020), with their vast parameter spaces and diverse, richly curated training datasets, are rapidly emerging as powerful tools within the finance sector. These models demonstrate exceptional capabilities in natural language processing (NLP) tasks and excel at analyzing extensive volumes of financial data, including news, investor communications, and regulatory reports (Wu et al., 2023). In the domain of finance, researchers have begun harnessing the potential of LLMs to enhance decision-making processes. For example, Lopez-Lira & Tang (2023) utilized ChatGPT to conduct sentiment analysis on news headlines, enhancing decision-making in stock trading. By delivering in-depth insights, conducting comprehensive risk assessments, and supporting investment decisions, LLMs are becoming integral components. However, their application within finance presents distinct challenges, particularly the necessity for extreme precision and reliability, given the specialized and high-risk nature of financial data. Current research focuses on overcoming these hurdles by refining algorithms, utilizing domain-specific training data, and integrating expert-driven systems. Despite these challenges, LLMs are uniquely positioned to augment and enhance algorithmic trading strategies, offering new opportunities for innovation in financial markets (Wu et al., 2023; Zhao et al., 2024).

Despite the growing interest in LLMs, their application within the realm of algorithmic trading remained underexplored. In this study, we meticulously examined six distinct LLMs, including LLaMA-2 (Touvron et al., 2023), LLaMA-3 (Touvron et al., 2023), Mistral-7B (Jiang et al., 2023), Falcon-7B (Almazrouei et al., 2023), OpenELM (Mehta et al., 2024), and OpenAI's latest model, GPT-4o (OpenAI et al., 2024). These LLMs serve as proxies for Deep Reinforcement Learning (DRL) methods by integrating real-world news to dynamically adjust trading agents actions based on LLM-driven insights. Furthermore, we introduced a comprehensive NLP-based fine-tuning methodology that empowers LLMs to emulate human-like trading decisions. The empirical validation of this work extends beyond the performance of LLMs, as we conducted a detailed study using stocks from `Silver` and `JPMorgan`. In summary, our contributions are as follows:

- First, we implemented Deep Q-Network (DQN) and Double Deep Q-Network (DDQN) based RL agents for algorithmic trading, utilizing a widely recognized algorithm.

- Second, we integrated LLMs, incorporating stock-related news to function as a proxy for modulating the behavior of the DRL agents by leveraging decisions made by the LLMs. This integration allowed for empirical validation of LLMs in algorithmic trading.

- Third, we introduced a novel instruction generation algorithm, termed `Stock-Evol-Instruct`, specifically designed for generating NLP datasets tailored to stock market forecasting. This algorithm dynamically adapts instructions based on historical financial data, market trends, and news, enabling the creation of datasets that are better aligned with real-world market conditions.

- Fourth, we fine-tuned two open-source LLMs, Mistral-7B, and LLaMA-3-8B, to function as fully LLM-based trading agents. These agents were fine-tuned using the innovative strategy developed in this study to emulate human-like trading decisions in the market.

## 2 RELATED WORK

RL in portfolio management often faces challenges such as poor generalization, market impact neglect, and inadequate consideration of causal relationships. To mitigate these issues, Kuo et al. (2021) developed the LOB-GAN generative model, which simulates a financial market to create a realistic training environment for RL agents, enhancing out-of-sample portfolio performance by

4%. Lussange et al. (2021) designed a multi-agent system (MAS) stock market simulator using RL, calibrated with London Stock Exchange data (2007-2018), which accurately replicates key market metrics, demonstrating effective agent learning. Furthermore, Pendharkar & Cusatis (2018b) explored intelligent agents for retirement portfolio management, revealing that an adaptive learning $TD(\lambda)$ agent outperformed traditional assets in cumulative returns. The shift toward algorithmic trading leveraging DRL is evident in the work of Zhou et al. (2024), who introduced a reward-driven DDQN algorithm incorporating human feedback, achieving up to 1502% cumulative returns across six datasets. Similarly, Huang et al. (2024) presented Multi-Agent Double Deep Q-Network (MADDQN) for balancing risk and return in financial trading, achieving a 23.08% average cumulative return. In stock prediction, Awad et al. (2023) combined DRL with ANN, LSTM, and SVMs, utilizing historical data and social media analysis to enhance prediction accuracy. Later, Taylor & Ng (2024) applied transformer models like BERT for stock price predictions, focusing on percentage changes derived from news articles.

Recent advancements in financial decision-making have highlighted the transformative potential of LLM-based frameworks. FinGPT (Yang et al., 2023) is an open-source LLM for the finance sector, where it takes a data-centric approach, providing researchers and practitioners with accessible and transparent resources. Summarize-Explain-Predict (SEP) (Koa et al., 2024) framework introduces explainable stock predictions, utilizing a self-reflective agent and Proximal Policy Optimization (PPO) to surpass traditional methods in accuracy and portfolio construction. Yu et al. (2023a) examined LLMs for forecasting NASDAQ-100 stocks, demonstrating their superiority over traditional models and emphasizing effective, explainable forecasts. FinMem (Yu et al., 2023b) introduces an agent with Profiling, Memory, and Decision-making components, enabling personalized, interpretable, and adaptable trading strategies. FinCon (Yu et al., 2024) models investment firm hierarchies through LLM-driven manager-analyst interactions and robust risk control mechanisms, excelling in stock trading and portfolio tasks. LEVER (Yuan et al., 2024b) uses an adaptive learning framework for high-frequency trading, integrating encoder-decoder architectures and active-meta learning for superior tick-level predictions. SePaL (Yuan et al., 2021) enhances dynamic corporate profiling via self-supervised learning on event graphs, yielding robust representations for financial tasks. Meanwhile, FinRL (Liu et al., 2022) democratizes quantitative finance through a DRL library with reproducible workflows for trading strategy development. News-driven approaches according to the review by Ashtiani & Raahemi (2023b) which they synthesized 61 studies from 2015-2022 on machine learning and text mining, noting the underexplored potential of news data compared to social media, despite its importance in financial predictions. In this manner, LLMFactor (Wang et al., 2024a) and MarketSenseAI (Fatouros et al., 2024) exploit LLMs for interpreting financial news and macroeconomic factors, achieving notable gains in market prediction. Reflection-driven systems like FinAgent (Zhang et al., 2024) leverage multimodal inputs and adaptive reflection mechanisms for improved returns, while debate-driven methods (Li et al., 2023) enhance automated trading via inter-agent debates and self-reasoning processes, achieving state-of-the-art accuracy. Alpha-GPT 2.0 (Yuan et al., 2024a) enhances alpha discovery with a Human-in-the-Loop approach, integrating human insights into AI-driven investment research. The study done by Wang et al. (2024b) introduces the QuantAgent framework, a two-layer LLM for autonomous agents, demonstrating effective trading signal mining and improved forecasting accuracy. These contributions collectively underscore the efficacy of LLMs in reshaping financial trading and decision-making landscapes.

## 3 METHODOLOGY

Figure 1 provide a detailed overview of the proposed framework and the methodologies employed in the development of our LLM-driven algorithmic trading system. The primary objective of this study is to integrate DRL with LLMs to optimize stock trading decisions by leveraging both historical market data and real-time news. We begin by outlining the DRL methods that serve as the foundational model for our trading agents. Subsequently, we elaborate on the design of our trading environment, wherein stock data and technical indicators are utilized to guide decision-making processes. We then introduce the pivotal role of LLMs, which augment the trading agents' capabilities by analyzing financial news to generate actionable insights. To further enhance these capabilities, we propose an innovative instruction generation algorithm, `Stock-Evol-Instruct`, designed to produce high-quality datasets specifically for stock market forecasting tailored for NLP-based agents. The integration of NLP-driven fine-tuning with DRL methodologies ensures the agents' ability to make adaptive decisions within a dynamic financial landscape.

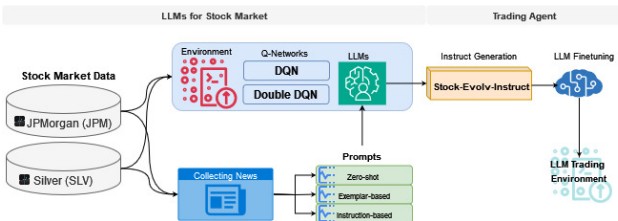

Figure 1: An overview of our proposed framework.

## 3.1 REINFORCEMENT LEARNING WITH DQN AND DDQN

DQN (Mnih et al., 2013; Park et al., 2024) and DDQN (Hasselt et al., 2016; Papageorgiou et al., 2024) are foundational algorithms in DRL, extensively utilized in sequential decision-making problems, particularly in the domain of algorithmic trading. The principal aim of the trading agent is to maximize long-term returns by selecting one of three possible actions: **buy**, **sell**, or **hold**, based on the current state of the market (environment). This decision-making scenario presents a dynamic and non-stationary environment that makes accurate predictions about future rewards a challenge. We have meticulously designed the DQN and DDQN algorithms, as specified within Appendix A/ Algorithm 1, with the primary objective of maximizing long-term rewards by selecting actions based on observed market conditions. The agent interacts with the stock market environment, aiming to learn an optimal trading policy over time.

In the **DQN** approach, a single Q-network $Q(s, a; \theta)$ is used to estimate the value of taking action $a$ in the current market state $s$, where $\theta$ represents the network's parameters. The agent updates its Q-values based on the observed rewards and transitions using the target function of $y_t = r_t + \gamma \max_{a'} Q(s_{t+1}, a'; \theta)$, where $y_t$ is the target Q-value, $r_t$ is the reward at time $t$, $\gamma \in [0, 1]$ is the discount factor, and $s_{t+1}$ is the next market state. The max operator selects the best action using the same network, which can lead to overestimation of Q-values. To address overestimation of Q-values in DQN, the **DDQN** algorithm employs two separate Q-networks: a *primary* Q-network $Q(s, a; \theta)$ and a *target* Q-network $Q_{\text{ast}}(s, a; \theta')$. The primary network is used for action selection, while the target network provides more stable target values during training. This decoupling helps the agent make better estimates of future rewards. The DDQN update uses the following target function:

$$y_t = r_t + \gamma \, Q_{\text{ast}} \left( s_{t+1}, \arg \max_{a'} Q(s_{t+1}, a'; \theta); \theta' \right)$$

Here, $\theta$ and $\theta'$ represent the parameters of the primary and target networks, respectively. By using the primary network to select the action $a'$, and the target network to evaluate it, DDQN reduces the overestimation of Q-values found in DQN. At each time step, the agent observes the current market state $s_t$, which may include historical price movements, technical indicators, and other market features. The agent selects an action using an $\epsilon$-greedy policy with a probability of $\epsilon$, the agent explores by selecting a random action, and with probability $1 - \epsilon$, it exploits its learned policy by selecting the action that maximizes the Q-value $a_t = \arg \max_a Q(s_t, a; \theta)$, where $a_t$ is the action that maximizes the expected reward, as predicted by the primary Q-network. Over time, $\epsilon$ decays, allowing the agent to shift from exploration to exploitation. After taking action $a_t$, the agent transitions to the next state $s_{t+1}$ and receives an immediate reward $r_t$. The experience $(s_t, a_t, r_t, s_{t+1})$ is stored in a replay memory buffer $\mathcal{M}$, which holds a fixed number of past experiences. Random sampling from this buffer during training reduces correlations between consecutive transitions, improving learning stability. At each training step, the agent samples a mini-batch of experiences from the replay buffer and updates the primary Q-network using the following loss function:

$$\mathcal{L}(\theta) = \mathbb{E}_{(s,a,r,s')} \left[ (y_t - Q(s, a; \theta))^2 \right]$$

where $y_t$ is the target Q-value for the current state and action, defined differently for DQN and DDQN. In both cases, the target network parameters $\theta'$ are periodically updated by copying the primary network's parameters: $\theta' \leftarrow \theta$. This update helps ensure that the target network provides stable estimates of future rewards, avoiding rapid fluctuations in the Q-value predictions. As a result, the agent learns a policy that maximizes long-term cumulative rewards by navigating the dynamic and uncertain stock market environment over time.

## 3.2 TRADING ENVIRONMENT

The trading environment algorithm, as described in Appendix A/Algorithm 2, is meticulously designed to facilitate robust decision-making within a trading context by dynamically determining the optimal action based on market conditions. Initially, the algorithm establishes the trading environment with a starting balance of $10,000$, zero shares, and zero profit, setting the stage for strategic engagement. For each action, the algorithm calculates critical performance indicators, including the current price difference (PD) and the moving average difference (MA). Depending on the agent's decision, the algorithm updates both the balance and profit accordingly. In scenarios where the agent opts to buy, the algorithm checks whether sufficient funds are available to acquire at least one share, updates the balance accordingly, and computes profit based on subsequent price movements. Conversely, when a selling action is executed, the algorithm updates the balance and profit based on the quantity of shares held and the market price at the time of the transaction.

In scenarios where an LLM is integrated, the algorithm substantially enhances decision-making processes by incorporating additional contextual information and strategic insights provided by the LLM. Specifically, when the LLM suggests a favorable action, the reward structure is dynamically adjusted to reflect this expert guidance. For instance, if the agent's action is to buy and the LLM corroborates this decision, the reward is doubled. Similarly, in cases where the LLM recommends a selling action, the reward is amplified by a factor of two, thereby reinforcing the decision-making framework with high-stakes endorsements. If the LLM advises the agent to hold, a fixed positive reward is allocated, promoting caution in volatile market conditions. In contrast, when the agent's action diverges from the LLM's suggestions, the reward is calculated based on price indicators, with penalties applied for invalid actions to deter poor trading behaviors. To ensure rewards remain within a reasonable range and prevent extreme fluctuations, all reward values are clipped to a defined range of -1 to 1. This strategic framework adeptly balances the reliance on technical indicators with the insights offered by the LLM, ultimately striving to optimize trading decisions.

## 3.3 NEWS ANALYTICS WITH LLMS FOR TRADING

The primary objective of this study is to forecast stock market actions based on the analysis of news headlines. We strategically leverage LLMs to interpret the news related to specific stocks and ascertain the optimal action to take. To achieve this, we employed news data from the Financial News Dataset available on Hugging Face (ashraq). This dataset provides real-world stock-related headlines pertinent to our case-study stocks, namely SLV and JPM, serving as the foundational input for the LLMs utilized in our experiments, facilitating a robust analysis of market sentiment. Our methodological approach encompasses a diverse range of prompting techniques, including zero-shot learning, instruction-following, and exemplar-based prompting, thereby enabling the LLMs to effectively adapt to various analytical contexts and extract actionable insights from the news.

### 3.3.1 PROMPT DESIGN

We utilized three prompt templates including (which are presented in Appendix B): **Prompt 1 – Zero-shot Forecasting**, this template presents the LLM with minimal context. It asks the model to make a decision on whether to buy, sell, or hold a stock based solely on the headline, without any additional guidance. **Prompt 2 – Instruction-based Forecasting**, building on zero-shot forecasting, this prompt includes additional instructions to guide the LLM toward better decision-making. It emphasizes the importance of sentiment analysis and the need to ignore irrelevant headlines. **Prompt 3 – Exemplar-based Forecasting**, where human-annotated examples are introduced. These exemplars provide the LLM with prior cases of buy, sell, or hold decisions based on similar headlines, enhancing its ability to generalize to new data.

### 3.3.2 LLMS FOR TRADING PREDICTIONS

To test these prompts, we selected a range of LLMs based on their ability to handle zero-shot and few-shot prompting effectively. We focused on a limited set of models that vary in size and training processes, ensuring diversity in architecture and capabilities. The selected models are OpenAI model GPT-4o (OpenAI et al., 2024), Meta LLMs such as LLaMA-2-7B (Touvron et al., 2023) and LLaMA-3-8B (Touvron et al., 2023), Mistral-7B (Jiang et al., 2023), Falcon-7B (Almazrouei et al., 2023), and OpenELM (Mehta et al., 2024). Per stock and per LLM, we generated three distinct

predictions using the designed prompts. These predictions were then integrated as signals within the trading environment to the DQN and DDQN trading algorithm, enabling the system to make informed decisions based on both historical market data and news.

## 3.4 TRADING AGENT

The objective of the trading agent is to create a fully NLP-based system that interprets market time-series data and news signals to make informed decisions regarding stock trading. The agent interacts with users and makes decisions based on stock market movements and news forecasts. To fine-tune LLM for this task, we developed a multi-step process that involved the generation of high-quality prompt templates, rating their quality, and evolving them under `Stock-Evol-Instruct` method to enhance the model's performance.

### 3.4.1 PROMPT GENERATION

We began by designing a series of prompt templates that direct the LLM to generate instructions for making trading decisions. The prompt template to generate a set of instruction templates is based on the following criteria: **(1) Price Movement**, comparing today's opening and closing prices and examining how the closing price aligns with the 2-day moving average. **(2) News Forecast**, incorporating news sentiment as a factor influencing the trading decision. The prompt template was designed to generate instruction in an emphasized decision-making process that considered the price difference between the day's open and close, a comparison between today's closing price and the 2-MA, and whether the news forecast supported an action.

The prompt generation template is presented in Appendix C. This approach resulted in 20 variant prompts designed for stock market forecasting. We specifically prompted LLM to consider variant *themes* while generating prompts. The themes are the central focus of each prompt and their intent in addressing trading decision-making. These themes help categorize the various types of decisions the trading agent may need to make, depending on the data and market signals it encounters. An example of a generated prompt and its theme is also presented in Appendix C.

However, we observe that prompts that are being generated may not be well-suited for the task, so expert judgment is required. Inspired by the LLM-as-a-Judge framework (Zheng et al., 2023), we employed LLMs once again to assess the quality of the prompt templates. Using a rating system from 1 to 100, the models evaluated the prompts based on how well they adhered to the task criteria. A rating of 50 indicated a neutral assessment, with scores above 50 representing increasingly higher quality and scores below 50 indicating poorer alignment with the task requirements. The judge LLM instruction is presented in Appendix D. Once all the prompt templates were rated, we applied a threshold of 80 to identify only **9** high-quality prompts. Templates that met this threshold were considered for further use, ensuring that only the most reliable instructions were retained.

### 3.4.2 STOCK-BASED AUTOMATIC INSTRUCTION DATA EVOLUTION

Inspired by the Evol-Instruct (Xu et al., 2024) method proposed by WizardLM and also used in WizardCoder (Luo et al., 2023), this work attempts to make stock market instructions to enhance the fine-tuning effectiveness of stock market-based agent that uses LLMs. The obtained 9 prompt templates within the prompt generation step, were used to generate an initial dataset which was later used for instruction evolution. The proposed `Stock-Evol-Instruct` (The visualization is presented in Figure 2) method is based on the original instruction evolution method which consists of three steps: 1) instruction evolving, 2) response generation, and 3) elimination evolving, i.e., filtering instructions that fail to evolve. We adapt each step according to the stock market-based trading requirements to generate high-quality data for fine-tuning the LLMs.

**1) Instruction Evolving.** We found that instruction generation methods are more complex and difficult for the stock domain. Additionally, they can generate entirely new instructions that are complex and do not match the task requirements. We initiate the instruction pool with the given initial instruction dataset. The Instruction Evolver method at Evol-Instruc uses an LLM to evolve instructions with two types: in-depth evolving and in-breadth evolving. We adapt each type specifically for the trading instruction generations. We added two more fields to the instructions to consider the themes of the initial prompts and an example representing the initial prompt. In-Depth Evolving enhances

instructions by making them more complex and difficult through four types of prompts: (1) *Adding Constraints* to ensure compliance with market regulations, (2) *Depending* incorporates dependencies between market factors for better context, (3) *Concretizing* refines abstract concepts into actionable signals, and (4) *Increase Reasoning*, enhances multi-step decision-making. In-Breadth Evolving aims to make the prompts topic coverage more and increase overall dataset diversity. Detailed In-Dept/Breadth Evolving prompts and stages are outlined in Appendix E.

**2) Response Generation.** Instead of using LLMs for generating responses (as the original Evol-Instruct does), we used rule-based decisions for stock trading based on a combination of price movements and news forecasts and is being used as a ground truth for fine-tuning LLMs. We found this more suitable than relying on LLMs for the generation of responses since we have time-series data to calculate the ground truth. It first calculates the price difference between the stock's opening and closing prices for the day, then compares the closing price to the 2-day moving average to detect short-term trends. If the closing price is higher than the opening price and above the 2-day moving average, the algorithm suggests a buy signal, while a lower closing price and being below the average triggers a sell decision. If no clear trend is identified, the default decision is to hold.

**3) Elimination Evolving.** We observed that evolved instructions contain unwanted information such as unknown placeholders, that is required to be entered within the prompts. This is a result of the LLM hallucinations and investigation of it deviates from the objective of the work. So we only eliminate those instructions that contain placeholders for auxiliary values. Also, we eliminated instructions that contain punctuation and stop words.

### 3.4.3 FINETUNING THE LLM ON THE STOCK EVOLVED INSTRUCTIONS

Once the prompt evolution process was complete, we used the Appendix F prefix which consists of time-series data to finalize the newly generated data for fine-tuning. During the finetuning RL agent, the instructions have access to the LLM outputs for news signals rather than news itself. Later, the dataset was shuffled, allowing the model to train on prompts of varying difficulty levels. By concatenating the finalized prompt with the rule-based decision at the prompt, the model was trained to generate appropriate trading decisions in a supervised manner. Our approach, inspired by recent advancements in instruction-tuning methods (Wang et al., 2023; Shin et al., 2020), ensures that the fine-tuned LLM is robust enough to handle the intricacies of stock market decision-making. This NLP-based trading agent can effectively analyze both numerical stock data and text-based news forecasts to guide users in making informed trading choices.

## 4 RESULTS

### 4.1 EXPERIMENTAL SETUPS

#### 4.1.1 METRICS

In the literature, two common metrics used as performance criteria are portfolio returns and differential SR (Pendharkar & Cusatis, 2018a). **Return on Investment (ROI),** that measures the profitability of an investment relative to its cost. It indicates how much return is generated for each dollar invested, making it useful for comparing different investments. A positive ROI means the investment is profitable, while a negative ROI indicates a loss. The formula for ROI is: $ROI = \frac{\text{Net Profit}}{\text{Investment Cost}} \times 100$. **Sharpe Ratio (SR)**, where it evaluates the risk-adjusted return of an investment or trading strategy by comparing the excess return (over the risk-free rate) to the investment's volatility (Bertoluzzo & Corazza, 2014). A higher SR implies better risk-adjusted performance, helping investors assess whether the returns justify the risks taken. The formula for the SR is: $SR = \frac{\text{Return} - \text{Risk-Free Rate}}{\text{Standard Deviation of Returns}}$.

#### 4.1.2 DATASETS

**Stock Time Series and News Data.** We utilize stock time series data for financial analysis, which includes historical stock prices. The stock data is collected daily, while the corresponding news articles are gathered from financial news sources. This dataset is crucial for training our trading agent and experimentation on LLMs, which make decisions based on both stock price movements and news. The large-scale nature of the work required more resources for experimentation, so we were convinced to use only these two stocks based on the number of available news articles in the

given large-scale (1.85M) financial news dataset (ashraq). So, we selected two representative stocks from different sectors—Silver (SLV) and JPMorgan (JPM)—to examine how the market reacts to different types of information. The Table 1 at Appendix G presents an overview of the period covered, the number of trading days, and the corresponding number of financial news articles for each stock. The time series data includes the opening, closing, high, and low prices for each trading day, while the news data provides new headlines for a given stock.

**Trading Agents Train Test Splits.** For training and evaluating our stock market trading agents, we split the stock time series and news data into train and test sets. This split is necessary to ensure that the models are tested on unseen data, simulating real-world trading conditions. The train set is used to finetune the trading agent, while the test set is used to evaluate its performance. Appendix G/Table 2 provides an overview of the train-test splits for both Silver (SLV) and JPMorgan (JPM). The data is categorized into three action classes: *Buy*, *Sell*, and *Hold*, each representing possible trading decisions. The total number of decisions for each split indicates the distribution of action labels across the dataset. These train-test splits ensure a balanced representation of each class, allowing our trading agents to learn and generalize across different market conditions. The alignment of stock price data with news is maintained throughout the splits, enabling the models to predict trading decisions based on both price movements and relevant financial news.

### 4.1.3 BASELINES

As baseline models, we used FinGPT (Yang et al., 2023) and FinRL (Liu et al., 2022) models. FinGPT is an open-source LLM that takes a data-centric approach to fine-tuning to provide researchers and practitioners an accessible and transparent resources for developing a financial LLM. This baseline model used for Q-Learning models over stock market data and trading agent backbone for comparison of the agent performance on real-world test data. Moreover, FinRL, is a DRL reproducible workflow for developing trading strategies. It provides virtual environments, real-world constraints, and advanced DRL algorithms such as DDPG, TD3, A2C, SAC, and PPO models. We tried with all five algorithms and only reported the best models per stock as the FinRL model result.

### 4.2 Q-LEARNING AND LLMS EVALUATIONS ON STOCK MARKET DATA

The evaluation of Q-learning models and LLMs on stock market data (Table 3, Appendix H) highlights their performance in trading decisions, showcasing strengths and weaknesses across prompts and architectures for Silver (SLV) and JPMorgan (JPM) stocks. In FinRL backend DRL models, for JPM we obtained better results with PPO, and for SLV we achieved better results with TD3 model.

**LLMs Outperforming Traditional RL Alone.** The integration of LLMs with RL models frequently outperformed traditional RL models alone. For example, the GPT-4o model, when paired with DDQN, achieved an SR of 2.43 for SLV using Prompt-2, demonstrating a clear advantage over the RL models. This suggests that leveraging LLMs can enhance decision-making processes in trading strategies, particularly by providing richer contextual understanding and better adapting to market signals. The substantial performance gains illustrate the potential of LLMs to not only augment traditional RL frameworks but also to redefine approaches to algorithmic trading.

**Inconsistencies in ROI.** While some combinations of LLMs and RL models achieved impressive SR, their corresponding ROI figures were not consistently positive. For instance, Mistral-7B with DDQN produced a Sharpe Ratio of 2.29 for JPM but resulted in a negative ROI of -10.39. This was also observed in some baseline models, where high SR did not always correlate with positive ROI. This discrepancy highlights the importance of not solely relying on SR as a performance indicator; ROI provides essential context regarding the overall profitability of the trading strategies. The findings indicate that while a high SR may suggest effective risk-adjusted returns, it does not guarantee overall profitability, thus necessitating a balanced evaluation of both metrics.

**Performance Variability Across Models.** The performance of LLMs in conjunction with RL models varied significantly depending on the prompt used. For instance, Prompt-2 yielded a notable improvement in the DDQN approach for the OpenELM-3B model when applied to the SLV stock, achieving an SR of 0.19 compared to the other prompts, which either produced negative SR values or lower ROI. This variability in performance was also observed in baseline models, highlighting the critical role of prompt selection in model performance. This indicates that certain prompts can effectively harness the strengths of specific models to improve trading decisions.

**Model and Prompt Synergy.** The results indicate a complex synergy between specific LLMs and prompts. The Falcon-7B model, for instance, performed well under various prompts, particularly with DDQN for both stocks, achieving a Sharpe Ratio of 2.18 for SLV with Prompt-2. In comparison, baseline models showed lower SRs, especially when applied to JPM. This suggests that certain model-prompt combinations can leverage market information more effectively, leading to better trading outcomes. Additionally, the interaction between the model architecture and prompt design indicates the need for iterative refinement and customization to align models with specific trading objectives and market conditions.

**Insights into Market Behavior.** The observed results also reflect broader market behavior and the capacity of LLMs to adapt to varying conditions. Models like GPT-4o and LLaMA-3-8B, which demonstrated potential for profitable trading decisions, may be better equipped to capture changes in market signals and sentiment shifts. This ability could be a crucial factor in navigating the complexities of stock trading, particularly in delicate environments where traditional models may hesitate.

The integration of LLMs with RL strategies shows promise for improving trading performance in stock market scenarios. While some models, particularly GPT-4o and LLaMA-3-8B, demonstrate significant potential for profitable trading decisions, others may require further refinement in prompt design and model training to enhance their effectiveness. The baseline results further underscore the importance of LLM-based models, especially in terms of SR. The findings underscore the importance of careful model selection and prompt engineering in developing robust trading agents capable of navigating complex financial markets. Fine-tuning LLMs with instruction datasets is essential for optimizing their performance in trading scenarios, as it allows the models to better understand specific tasks and trading strategies. By providing tailored prompts that clarify objectives, the models can generate more accurate and actionable insights, leading to improved decision-making. This targeted training can significantly enhance performance, ultimately resulting in more effective trading agents capable of navigating complex financial markets.

### 4.3 Trading Agent Results on Real World Test Split

The fine-tuned trading agents, tested on instruction datasets and real-world splits, showed competitive performance across JPM and SLV. The results are presented in Table 4 (Appendix H).

For JPM, LLaMA-3 achieved an F1-Score of 81.53, driven by high recall of 86.23% and precision of 84.87%, indicating a strong ability to identify profitable trades with minimal false positives. Despite this, its ROI of 23.78%, while positive, is lower compared to Mistral-7B's ROI of 53.15%. Mistral-7B, though it exhibited lower precision and recall (74.33% and 71.31%), seems to have identified fewer but more profitable trades, leading to a much higher overall return. This suggests that while LLaMA-3 has a higher accuracy in trade prediction, Mistral-7B may excel in optimizing financial outcomes under real-world conditions. On the SLV stock, both models showed improved performance, with Mistral-7B outperforming LLaMA-3 in terms of both recall (87.01% vs. 85.81%) and F1-Score (78.01 vs. 75.88). Interestingly, Mistral-7B precision improved significantly to 80.36%, indicating a more balanced ability to correctly predict trades while maintaining profitability. The ROI for SLV was strong for both models, with LLaMA-3 achieving 44.93% and Mistral-7B slightly outperforming at 48.36%, highlighting the strength of Mistral-7B's fine-tuning.

Baseline models provided additional context for evaluating the fine-tuned agents. FinRL (Liu et al., 2022), a fully RL-based model, yielded a minimal ROI of 0.04% for JPM and 7.33% for SLV, indicating limited profitability when relying solely on stock time-series data. FinGPT (Yang et al., 2023), a fine-tuned model on trading datasets, exhibited a negative ROI of -8.28% for JPM and -20.58% for SLV, demonstrating challenges in leveraging natural language data for consistent profitability. Results demonstrate the value of fine-tuning LLMs with instruction datasets for trading tasks. LLaMA-3's stronger F1-Scores highlight its consistent performance in predicting trades, while Mistral-7B's higher ROI across both stocks underscores its ability to translate predictions into real profits. The comparison with baseline models underscores the significant advancements achieved through fine-tuning, highlighting the limitations of traditional RL and open-source FinGPT models in real-world trading environments. The combination of fine-tuned instruction data and real-world test environments allows these LLMs to operate as robust trading agents, each with distinct strengths in different market scenarios.

## 5 DISCUSSIONS AND LIMITATIONS

**Role of Prompt Design with LLMs.** The analysis of the results presented in Table 3 points to the critical role of prompt design in optimizing the performance of LLMs in trading applications. The variability in SR and ROI across different prompts and models suggests that tailored prompts can significantly influence the effectiveness of LLMs in conjunction with RL strategies.

**Nature of LLMs.** LLMs showed a great capability in handling domain-specific tasks such as stock market trading. The fine-tuning process revealed that while LLMs like GPT-4o, LLaMA, and Mistral excel at synthesizing news-based insights, their performance in trading scenarios still benefits from instruction-based finetuning improvements with agent-based modeling. By refining the instruction sets through our `Stock-Evol-Instruct` framework, we could observe measurable gains in decision-making. This can be beneficial for even a domain-specific fine-tuned LLM, such as FinGPT to enhance decision-making capabilities. While domain-specific LLMs are often tailored to specific financial contexts, integrating the instruction methodology introduced in this work could significantly improve their performance.

**Limited number of Generated Prompts.** One of the key considerations in this study is the limited number of generated prompts used for instruction generation, with only 20 prompts designed to simulate real-world trading scenarios. While a larger set of prompts could have potentially enhanced the diversity of trading decisions, we intentionally restricted the scope as the primary objective was to assess and fine-tune LLMs within a controlled environment. Expanding the number of prompts, especially by leveraging various LLMs for prompt generation, could offer further insights into how models adapt to more intricate market situations, presenting a promising direction for future work.

**Inconsistency between SR and ROI:** The inconsistency between SR and ROI can arise from several factors. While SR measures risk-adjusted returns, it may not fully reflect the overall profitability of a trading strategy, particularly in conditions where high risk can lead to significant gains or losses. But, ROI focuses solely on profitability, ignoring risk, which can cause variations when models perform well in risk-adjusted terms (high SR) but fail to achieve consistent positive returns (low ROI), especially if they are not effectively utilizing market opportunities or if their risk exposure is skewed with profitability. According to Table 3, the choice of model and prompt can contribute to these inconsistencies. As observed, some LLMs, such as GPT-4o or LLaMA-3, exhibit high ROI but struggle to manage risk effectively, resulting in lower SR. In contrast, certain RL models like DDQN may display lower ROI but demonstrate more stable performance, leading to higher SR. Additionally, variations across different prompts (PT-1, PT-2, PT-3) can introduce changes in model behavior that impact ROI and SR differently. While some prompts enhance ROI by capturing market trends, they may increase fluctuations, thus lowering the SR. In most scenarios, Prompt 3 (PT-3), which uses examples to guide models, improves the SR but leads to significantly lower ROI.

## 6 CONCLUSIONS

The integration of LLMs into algorithmic trading represents a promising advancement in enhancing decision-making processes within dynamic financial environments. As financial markets grow increasingly intricate and volatile, the capability of LLMs to deliver dynamic, human-like trading strategies presents new opportunities for innovation within the finance industry. Research aimed at expanding the instruction generation process and applying LLMs to a wider array of market scenarios could significantly enhance their applicability and reliability in high-stakes trading environments. By leveraging both RL techniques and LLM-driven insights from real-time financial news, this study demonstrated that fine-tuned models, such as LLaMA and Mistral, can effectively serve as trading agents, making human-like decisions in response to market changes. Moreover, the proposed `Stock-Evol-Instruct` method, meticulously crafted for generating high-quality instruction datasets, enhances the performance of LLMs by adapting to the complexities of real-world stock market conditions. Through rigorous empirical validation on case-study stocks, including SLV and JPM, our results demonstrate that the incorporation of NLP-driven models into RL-based trading systems not only improves the accuracy of trading agents but also enhances their adaptability, paving the way for more sophisticated trading strategies in the future.

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

# A ALGORITHMS

**Algorithm 1** DQN and DDQN Algorithm

1: **Initialize**: $Q(s, a; \theta)$                                                   ▷ policy network

2: **Initialize**: $Q_{\text{ast}}(s, a; \theta')$                                                ▷ target network

3: **Input**: $N_{epoch}, N_{memory}, N_{step}, batch\_size, freq_{train}, Q_{update\_freq}$

4: **Set**: Experience replay memory $\mathcal{M}$ with capacity $N$

5: **Set**: $\epsilon = 1.0, \epsilon_{\min} = 0.1, \epsilon_{\text{decrease}} = 1e - 3, \gamma = 0.97$

6: **for** $e = 1$ to $N_{epoch}$ **do**

7:     **Initialize**: $s_0 \leftarrow$ env.reset(), $t \leftarrow 0$, total\_reward $\leftarrow 0$, total\_loss $\leftarrow 0$

8:     **while** $t < N_{step}$ **do**

9:         **Action selection via $\epsilon$-greedy:**

10:         **if** Random Action $> \epsilon$ **then**

11:             $a_t \leftarrow$ Random Action

12:         **else**

13:             $a_t \leftarrow \arg\max_a Q(s_t, a; \theta)$

14:         **end if**

15:         **Environment step:** $(s_{t+1}, r_t) \leftarrow$ env.step$(a_t)$

16:         **Store transition in memory:** $\mathcal{M} \leftarrow \mathcal{M} \cup (s_t, a_t, r_t, s_{t+1})$

17:         **if** $|\mathcal{M}| > N$ **then**

18:             Remove the oldest transition from $\mathcal{M}$

19:         **end if**

20:         **if** $|\mathcal{M}| = N$ and $t \mod freq_{train} = 0$ **then**

21:             Sample a minibatch of size $batch\_size$ from $\mathcal{M}$

22:             **for** each sampled transition $(s, a, r, s')$ **do**

23:                 **if** $s'$ is terminal **then**

24:                     $y_i = r_i$

25:                 **else**

26:                     **if** Using DDQN **then**

27:                         $y_i = r_i + \gamma \, Q_{\text{ast}}(s', \arg\max_a Q(s', a; \theta); \theta')$

28:                     **else**

29:                         $y_i = r_i + \gamma \max_a Q(s', a; \theta)$

30:                     **end if**

31:                 **end if**

32:             **end for**

33:             Update $\theta$ by minimizing loss: $L(\theta) = \frac{1}{\text{batch\_size}} \sum_i (y_i - Q(s_i, a_i; \theta))^2$

34:         **end if**

35:         **if** $t \mod Q_{update\_freq} = 0$ **then**

36:             Update target network: $\theta' \leftarrow \theta$

37:         **end if**

38:         **if** $\epsilon > \epsilon_{\min}$ and $t > $ start\_reduce\_epsilon **then**

39:             $\epsilon \leftarrow \max(\epsilon - \epsilon_{\text{decrease}}, \epsilon_{\min})$

40:         **end if**

41:         $s_t \leftarrow s_{t+1}, t \leftarrow t + 1$

42:     **end while**

43: **end for**

**Algorithm 2** Trading Environment

1: **Initialize**: balance $B \leftarrow 10000$, profit $PF \leftarrow 0$, shares $SH \leftarrow 0$, Reward $R \leftarrow 0$
2: **Input:** action $a_t \in \{\text{buy, sell, hold}\}$
3: **Calculate**: $PD \leftarrow$ current price difference, $MA \leftarrow$ moving average difference
4: **if** $a_t$ is buy **then**
5:      $SH_{to\_buy} \leftarrow \frac{B}{Price_{open}}$
6:      **if** $SH_{to\_buy} \geq 1$ **then**
7:          $B \leftarrow B - (Price_{open} \times SH_{to\_buy})$
8:          $PF \leftarrow PF + (Price_{close} - Price_{previous\_close}) \times SH_{to\_buy}$
9:          $SH \leftarrow SH + SH_{to\_buy}$
10:         $r' \leftarrow 100 \times \frac{PF}{B}$
11:         **if** LLM news says to buy **then**        ▷ This **if** is active, when LLM used
12:            $R \leftarrow r' \times 2$
13:         **else if** $PD \geq MA$ **then**
14:            $R \leftarrow r'$
15:         **end if**
16:      **else**
17:         $R \leftarrow R - 20$             ▷ Penalty reward for invalid buy
18:      **end if**
19: **else if** $a_t$ is sell **then**
20:      **if** $SH \neq 0$ **then**
21:         $PF \leftarrow (Price_{close} - Price_{open}) \times SH$
22:         $B \leftarrow B + (SH \times Price_{open})$
23:         $SH \leftarrow 0$
24:         $r' \leftarrow 100 \times \frac{PF}{B}$
25:         **if** LLM news says to sell **then**       ▷ This **if** is active, when LLM used
26:            $R \leftarrow R \times 2$
27:         **else if** $PD \geq MA$ **then**
28:            $R \leftarrow r'$
29:         **end if**
30:      **end if**
31: **else if** $a_t$ is hold **then**
32:      **if** LLM news says to hold **then**       ▷ This **if** is active, when LLM used
33:         $R \leftarrow 2$
34:      **end if**
35: **end if**
36: $R \leftarrow \max(-1, \min(1, R))$             ▷ Clipping reward

## B PROMPT DESIGN

**Prompt 1: Zero-shot Forecasting.**

```
Given the following news headlines, determine whether to "buy", "sell",
    or "hold" that stock.
Notes:
- Output should only be "Buy," "Sell," or "Hold". No more explanation or
    additional text at output.

### Stock Name: {stock_name}
### Stock Code: {stock_code}
### News Headlines:
{headline}
### Prediction (Buy/Sell/Hold):
```

**Prompt 2: Instruction-based Forecasting.**

```
Stock Market Prediction Task: The task is to generate a decision on
    whether it is a good day to buy, sell, or hold a stock based on the
    news headlines.
Notes:
- The sentiment can be a good criterion to look and decide whether to buy
     that stock, sell it, or hold and do nothing.
- Ignore headlines that are not relevant to the defined stock.
- Output should only be "Buy," "Sell," or "Hold". No more explanation or
    additional text at output.

Given the following news headlines, determine whether to "buy", "sell,"
    or "hold" that stock.
### Stock Name: {stock_name}
### Stock Code: {stock_code}
### News Headlines:
{headline}
### Prediction (Buy/Sell/Hold):
```

**Prompt 3: Exemplar-based Forecasting.**

```
Stock Market Prediction Task: The task is to generate a decision on
    whether it is a good day to buy, sell, or hold a stock based on the
    news headlines.
Notes:
- The sentiment can be a good criterion to look and decide whether to buy
     that stock, sell it, or hold and do nothing.
- Ignore headlines that are not relevant to the defined stock.
- Output should only be "Buy," "Sell," or "Hold". No more explanation or
    additional text at output.

Examples:
<examples>

Given the following news headlines, determine whether to "buy", "sell,"
    or "hold" that stock.
### Stock Name: {stock_name}
### Stock Code: {stock_code}
### News Headlines:
{headline}
### Prediction (Buy/Sell/Hold):
```

## C PROMPT GENERATION

**Prompt Generation Template**

```
This is an instruction generation task. You should generate 20 different
    prompt templates based on the following information. You can use
    criteria inside of the prompt template.

<task-definition>
As a trading agent, they make buy, sell, or hold decisions based on the
    statistical data provided for the previous and current trading days,
    as well as news forecasts.
</task-definition>

<task-criteria>
1. **Price Movement:**
   - Calculate the price difference as the difference between today's
       closing price and today's opening price.
   - Compare today's closing price with the 2-day moving average.
2. **News Forecast:**
   - Consider the news forecast for today as an additional factor
       influencing your decision.

**Instructions:**
1. Based on the price difference, compare today's closing price to the 2-
    day moving average and incorporate today's news forecast.
2. Decide whether to "buy", "sell", or "hold" based on:
   - The price difference between today's open and close.
   - Whether today's closing price is above or below the 2-day moving
       average.
   - The news forecast for today.

Make your decision by weighing these factors carefully.

Your final decision should be one of the following:
- **"Buy"** if the price movement indicates a strong upward trend and the
    news forecast supports this action.
- **"Sell"** if the price movement indicates a strong downward trend and
    the news forecast supports this action.
- **"Hold"** if the price movement is neutral or unclear, or if the news
    forecast suggests caution.
</task-criteria>

<input-data>
**Stock Info:**
- **Stock Name:** {stock_name}
- **Stock Code:** {stock_code}

**Previous Day's Statistics:**
- **Opening Price (Previous Day):** {p_open}
- **Highest Price (Previous Day):** {p_high}
- **Lowest Price (Previous Day):** {p_low}
- **Closing Price (Previous Day):** {p_close}
- **2-Day Moving Average (Previous Day):** {2_ma_diff}
- **News Forecast (Previous Day):** {p_news}

**Today's Statistics:**
- **Opening Price (Today):** {t_open}
- **News Forecast (Today):** {t_news}
</input-data>

The output should be a JSON in the following format
[
{"prompt-id": "Numeric ID", "name":"title for the instruction type", "
    theme": "theme of the prompt-template", "prompt-template":"prompt
    template"},
....
]
```

**An Example of Generated Prompt**

```
Theme: Consistent Downward Trend

Prompt Template: If today's closing price ({t_close}) shows a consistent
    downward trend compared to the opening price ({t_open}) but the news
    forecast ({t_news}) is positive, review the 2-day moving average ({2
    _ma_diff}) and consider 'holding' or 'buying' depending on additional
     factors.
```

## D    PROMPT GENERATION TEMPLATE RATING

We added the following prefix to the prompt generation template in addition to the generated prompts to rate the quality of prompts and provide an explanation for the ratings.

```
A prompt template was generated using task criteria, and now rate them
    based on the task criteria and input data.

- Please rate them on a scale from 1 to 100, where 1 represents the
    lowest quality and 100 represents the highest quality. A rating of 50
     is neutral, ratings between 50 and 100 indicate increasing levels of
     good to excellent value, and ratings from 1 to 50 indicate
    decreasing levels of quality.
- Add rating as a 'rating' key to the prompt dict.
- 'name' refers to the sub-category of the theme and it is an objective
    of the prompt template.
- The prompt templates try to follow the task criteria so you should rate
     based on the task criteria and prompt template quality on reflecting
     those criteria in the prompt template.
```

## E   INSTRUCTION EVOLVING

**Stock-Evol-Instruct.** After prompt generation, the instruction evolving technique uses in-breadth base instruction and in-depth base instructions to generate further five prompts using different evolutions. In in-breath evolving, it uses the same prompt template (the one obtained from Appendix C) with a filled example to generate a new prompt. Similarly in in-depth evolution, the same prompt template with a filled example is used to generate four new prompts using different objectives, such as adding constraints, depending, concretizing, and increasing reasoning to the prompts. Lastly, an elimination step looks for new prompts that don't contain valid placeholders. The process is visualized in Figure 2. In in-depth evolution, the four different evolutions are being considered with the following goals:

- **Add Constraints**: Introduce rules or limits based on market regulations. This ensures that trading strategies comply with requirements.
- **Depending**: Incorporate dependencies between market factors, such as news sentiment, or focus on specific into certain issues that can be beneficial in understanding the market.
- **Concretizing**: Refine high-level concepts into actionable signals, such as specific buy/sell thresholds or open/close conditions. This makes the strategies directly applicable to live trading scenarios and reduces ambiguity as it introduces more specific concepts rather than general ones.
- **Increase Reasoning**: Enhance the model's ability to interpret and react to complex market patterns by integrating multi-step reasoning.

**In-Depth Evolving Prompt**

```
I want you act as a Prompt Creator.
Your goal is to draw inspiration from the #Given Prompt# to create a
    brand new prompt.
```

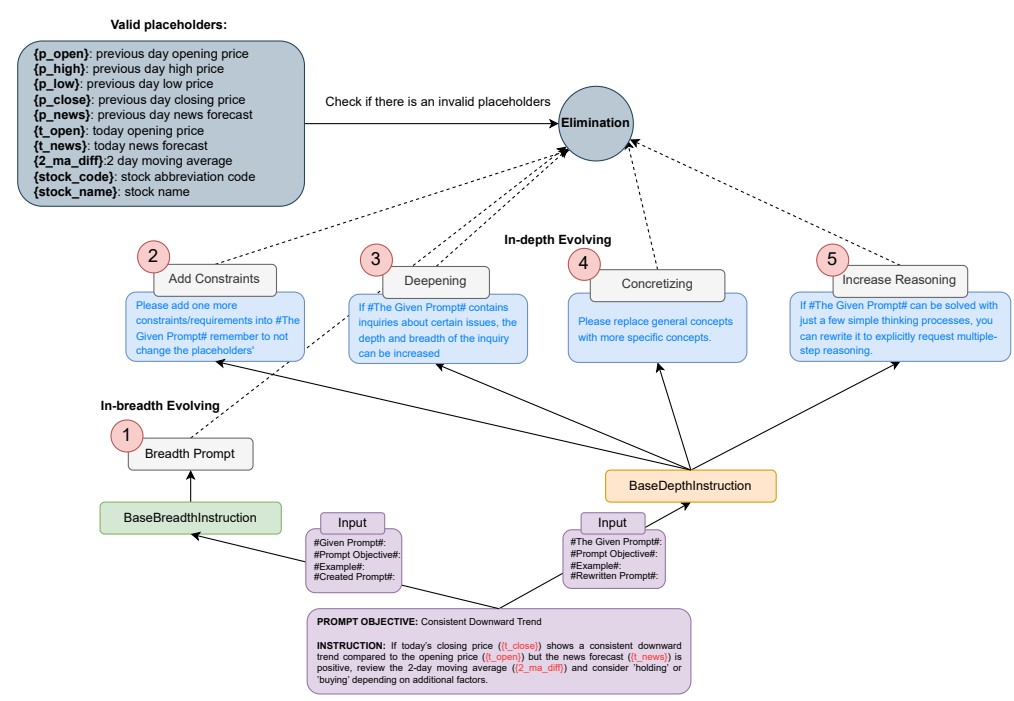

Figure 2: `Stock-Evol-Instruct`, a stock-based automatic instruction data evolution.

```
This new prompt should belong to the same domain as the #Given Prompt#
    but be even more rare.
The LENGTH and complexity of the #Created Prompt# should be similar to
    that of the #Given Prompt#.
The #Created Prompt# must be reasonable and must be understood and
    responded by humans.
We do not need models to provide explanation. So do not add asking
    explanations in prompts.
The prompts are sensitive to the stock name and code. Do not change them.
Placeholders will be given for the inputs, do not change them at any cost
As a Prompt Creator you will receive #Example# which consists of filled
    real world data into the prompt.
An objective of #Given Prompt# will be provided in #Prompt Objective# and
    the new prompt should follow the same objective.
```

**In-Breadth Evolving prompt**

```
I want you act as a Prompt Rewriter.
Your objective is to rewrite a given prompt into a more complex version
    to make those famous AI systems (e.g. GPT4) a bit harder to handle.
But the rewritten prompt must be reasonable and must be understood and
    responded by humans.
Your rewriting cannot omit the non-text parts such as the table and code
    in #The Given Prompt#:. Also, please do not omit the input in #The
    Given Prompt#.
We do not need models to provide explanation. So do not add asking
    explanations in prompts.
The prompts are sensitive to the stock name and code. Do not change them.
Placeholders will be given for the inputs, do not change it at any cost.
As a Prompt Creator you will receive #Example# which consists of filled
    real world data into the prompt.
```

```
An objective of #Given Prompt# will be provided in #Prompt Objective# and
    the new prompt should follow the same objective.
You should try your best not to make the #Rewritten Prompt# become
    verbose, #Rewritten Prompt# can only add 10 to 20 words into #The
    Given Prompt#."""
```

## F  FINETUNING THE LLM

```
Stock Market Prediction.
Given input stock market data, forecast today's action should be 'buy', '
    sell', or 'hold'.

**Stock Info:**
- **Stock Name:** {stock_name}
- **Stock Code:** {stock_code}

**Previous Day's Statistics:**
- **Opening Price (Previous Day):** {p_open}
- **Highest Price (Previous Day):** {p_high}
- **Lowest Price (Previous Day):** {p_low}
- **Closing Price (Previous Day):** {p_close}
- **2-Day Moving Average (Previous Day):** {2_ma_diff}
- **News Forecast (Previous Day):** {p_news}

**Today's Statistics:**
- **Opening Price (Today):** {t_open}
- **News Forecast (Today):** {t_news}
```

## G  DATASETS

Table 1: Stock time series and news statistics

| Stock Name | Stock Code | Start-Date | End-Date | No. of Days | No. of News |
|------------|-----------|------------|------------|------------|------------|
| Silver | SLV | 2018-12-27 | 2020-06-03 | 360 | 506 |
| JPMorgan | JPM | 2018-09-15 | 2020-06-03 | 430 | 554 |

Table 2: Trading agents train and test set statistics. The time-series data is split into train-test sets before the generation of instructions for building train and test sets.

| Stock Name | Stock Code | Split | Days | Buy | Sell | Hold | Total |
|------------|-----------|-------|------|-----|------|------|-------|
| Silver | SLV | Train | 249 | 609 | 85 | 361 | 1055 |
|  |  | Test | 106 | 195 | 29 | 53 | 277 |
| JPMorgan | JPM | Train | 298 | 810 | 198 | 288 | 1296 |
|  |  | Test | 127 | 167 | 63 | 83 | 313 |

## H  RESULTS

Table 3: Results of Q-learning and LLMs on stock market data. Per LLM evaluation we ran the RL model for fair comparison.

| Stock | Prompt | LLM | RL | | | | RL + LLM | | | |
| | | | DQN | | DDQN | | DQN | | DDQN | |
| | | | SR | ROI | SR | ROI | SR | ROI | SR | ROI |
|---|---|---|---|---|---|---|---|---|---|---|
| | PT-1 | | -0.09 | 17.01 | -0.33 | **17.61** | -0.58 | 17.72 | -0.61 | 17.31 |
| | PT-2 | FinGPT (Yang et al., 2023) | -0.10 | 17.60 | -0.18 | 17.25 | -1.29 | 16.56 | -1.35 | 16.57 |
| | PT-3 | | -0.30 | 16.97 | 0.12 | 16.45 | -1.07 | 17.13 | -1.33 | 16.81 |
| | PT-1 | | -0.36 | -5.06 | -0.66 | -4.95 | -1.67 | 16.60 | -1.50 | 16.86 |
| | PT-2 | OpenELM | -0.59 | -4.33 | **0.19** | -5.79 | -0.97 | **17.33** | -0.90 | 17.12 |
| | PT-3 | | 0.13 | -5.36 | -0.10 | -5.26 | -1.12 | 17.07 | -1.04 | 17.19 |
| | PT-1 | | -0.49 | -4.73 | 0.14 | -5.24 | -1.17 | 17.26 | -1.04 | 16.57 |
| | PT-2 | LLaMA-2 | 0.13 | -5.72 | -0.50 | -5.05 | -1.54 | 17.55 | -1.51 | 16.55 |
| | PT-3 | | -0.24 | -4.85 | **-0.25** | -5.02 | -1.55 | **17.78** | -1.37 | 17.36 |
| SLV | PT-1 | | -0.65 | -5.27 | -0.11 | -5.70 | -0.11 | 16.88 | -0.12 | 17.22 |
| | PT-2 | LLaMA-3 | -0.31 | -5.26 | -0.30 | -4.60 | 0.64 | 17.01 | **0.74** | **17.53** |
| | PT-3 | | -0.42 | -4.77 | 0.20 | -5.32 | -0.92 | 17.32 | -0.92 | 16.82 |
| | PT-1 | | -0.51 | -5.06 | -0.45 | -5.26 | 1.96 | 17.00 | 1.78 | 16.70 |
| | PT-2 | GPT-4o | -0.33 | -4.92 | 0.14 | -5.14 | **2.43** | **17.63** | 2.20 | 16.68 |
| | PT-3 | | 0.14 | -5.57 | 0.31 | -5.91 | 1.97 | 17.11 | 2.12 | 17.37 |
| | PT-1 | | -0.23 | -4.87 | -0.47 | -4.45 | -1.43 | **18.27** | -1.44 | 17.51 |
| | PT-2 | Falcon | 0.31 | -5.69 | 0.14 | -5.56 | 1.99 | 17.23 | **2.18** | **17.88** |
| | PT-3 | | 0.19 | -5.84 | -0.31 | -4.97 | -1.78 | 17.21 | -1.26 | 17.55 |
| | PT-1 | | -0.44 | -5.04 | -0.30 | -5.61 | -1.60 | 16.76 | -1.23 | 16.44 |
| | PT-2 | Mistral | -0.16 | -5.11 | 0.22 | -5.76 | 1.43 | 16.91 | **1.44** | 17.05 |
| | PT-3 | | -0.35 | -4.86 | -0.37 | -5.12 | 0.44 | 16.78 | 0.62 | **17.37** |
| | PT-1 | | 0.17 | -14.10 | -0.25 | -9.78 | 0.51 | -10.20 | 0.51 | -9.22 |
| | PT-2 | FinGPT (Yang et al., 2023) | -0.20 | -10.51 | -0.17 | -10.63 | -0.32 | -9.59 | -0.36 | -9.67 |
| | PT-3 | | 0.05 | -13.74 | -0.24 | -10.32 | 3.81 | -11.19 | 4.47 | -8.57 |
| | PT-1 | | -0.18 | -5.61 | -0.17 | -5.68 | -1.81 | -9.67 | -1.74 | -10.99 |
| | PT-2 | OpenELM | 0.24 | -8.68 | -0.20 | -5.21 | -1.51 | -10.68 | -1.48 | -9.21 |
| | PT-3 | | **0.29** | -7.09 | -0.26 | -5.57 | -1.52 | -10.58 | -1.60 | -10.08 |
| | PT-1 | | -0.28 | -5.38 | -0.51 | -5.73 | -1.53 | -10.01 | -1.68 | -9.66 |
| | PT-2 | Falcon | 0.19 | -7.12 | -0.29 | -4.74 | **1.95** | -10.29 | **2.19** | -9.26 |
| | PT-3 | | -0.29 | -6.04 | -0.19 | -4.75 | 1.52 | -10.74 | 1.81 | -10.11 |
| | PT-1 | | 0.29 | -8.30 | 0.24 | -8.54 | -0.94 | -11.57 | -0.92 | -9.85 |
| JPM | PT-2 | LLaMA-2 | -0.21 | -5.46 | -0.21 | -5.14 | -1.57 | -10.90 | -1.88 | -9.99 |
| | PT-3 | | -0.25 | -4.95 | -0.18 | -5.57 | **2.87** | -9.95 | **3.35** | -10.34 |
| | PT-1 | | -0.15 | -5.52 | 0.11 | -7.27 | **2.26** | -9.84 | **2.95** | -8.87 |
| | PT-2 | LLaMA-3 | -0.19 | -3.86 | 0.24 | -9.48 | 2.24 | -9.65 | 2.64 | -8.58 |
| | PT-3 | | -0.17 | -5.28 | -0.23 | -4.99 | 2.67 | -11.53 | 3.08 | -8.85 |
| | PT-1 | | -0.25 | -5.48 | 0.14 | -5.57 | -1.55 | -9.75 | -1.79 | -9.67 |
| | PT-2 | GPT-4o | -0.28 | -5.24 | -0.12 | -4.86 | 2.73 | -9.88 | 2.12 | -10.00 |
| | PT-3 | | 0.24 | -5.77 | 0.24 | -5.74 | -1.55 | -11.12 | -1.47 | -9.92 |

Table 4: Results of trading agents over instruction test dataset and real-world trading environment. The FinRL (Liu et al., 2022) is a fully RL agent that uses stock time-series data. FinRL supports models including DDPG, TD3, A2C, SAC, and PPO in the backend, for JPM we obtained better results with PPO, and SLV we obtained better results with TD3 models. FinGPT (Yang et al., 2023) is a finetuned open-source model over a trading dataset that operates on natural language text.

| Model | JPM | | | | SLV | | | |
| | Prec | Rec | F1 | ROI | Prec | Rec | F1 | ROI |
|---|---|---|---|---|---|---|---|---|
| *Baseline Models* | | | | | | | | |
| FinRL (Liu et al., 2022) | - | - | - | 0.04 | - | - | - | 7.33 |
| FinGPT (Yang et al., 2023) | 50.45 | 36.89 | 25.02 | -8.28 | 50.94 | 35.05 | 15.23 | -20.58 |
| *Proposed Models* | | | | | | | | |
| LLaMA-3-8B-Finetuned | 84.87 | 86.23 | 81.53 | 23.78 | 78.64 | 85.81 | 75.88 | 44.93 |
| Mistral-7B-Finetuned | 74.33 | 71.31 | 70.89 | 53.15 | 80.36 | 87.01 | 78.01 | 48.36 |

