# OpenReview forum: "Advancing Algorithmic Trading with Large Language Models: A Reinforcement Learning Approach for Stock Market Optimization"
_ICLR.cc/2025/Conference — Submitted to ICLR 2025_

### Official Review · Reviewer_MGL7 · 2024-10-21

**Soundness:** 2
**Presentation:** 1
**Contribution:** 2
**Rating:** 3
**Confidence:** 4

**Summary:**

This paper introduces a method leveraging Large Language Models (LLMs) to enhance reward mechanisms in a reinforcement learning (RL) trading algorithm, specifically a Double Deep Q-Network (DDQN). Traditionally, RL trading algorithms derive rewards solely from investment returns. However, this paper explores using LLMs to modify or replace conventional rewards in certain scenarios, aiming to help the RL agent trade more like a human. Additionally, the authors use prompt-based techniques to create a novel stock market dataset for fine-tuning the LLM. The paper evaluates the performance of different LLMs, using various prompts, on two stocks (JPMorgan and Silver), showing that RL agents incorporating LLM-generated rewards outperform those using traditional reward structures.

**Strengths:**

The authors tested a wide range of LLMs with various prompt types, and the writing is clear and concise in most sections.

**Weaknesses:**

It's standard practice in stock trading papers to show trading trajectories in charts, but this was missing here.
The paper lacks a clear justification for choosing JPMorgan and Silver for testing. Since LLMs rely heavily on textual data, similar studies typically select stocks based on the amount of news coverage during a given period.
While the paper evaluates each LLM and prompt, it fails to include common baselines like "buy and hold" strategies. Additionally, it does not compare results with a baseline using only LLMs for stock trading, which would help show the added value of the RL trading algorithm.
The relationship between the RL algorithm and LLM is difficult to follow. Often times, LLM or RL agents alone can be used for trading. Therefore, it is not obvious to user how does their combination work. This paper uses LLMs to refine rewards in addition to price-based rewards for the RL trading algorithm. This could be explained more clearly in both the text and accompanying figure (figure 1).

**Questions:**

During evaluation, does the RL agent have access to the news information or LLM output? Need to be more clear about what input features used during evaluation.

Could you include a visual representation of the trading trajectories in your experiments? It’s a common approach in stock trading papers and helps readers better understand the performance over time.

What was the rationale behind choosing JPMorgan and Silver for testing?

More detailed explanations in the text and clearer figures could improve the reader's understanding in the relationship between LLM and RL agent.

---

> ### Author Response · Authors · 2024-11-27
>
> Dear Reviewer,
>
> We sincerely thank you for all the effort and time you have dedicated to reviewing our work.
>
> 1.	During the evaluation, the RL agent, which uses instruction has access to the LLM outputs for news-based forecasting.
> 2.	We appreciate your suggestion for a visual representation of the trading trajectories, as we agree it would enhance the readability of the paper. However, due to the extensive nature of the experiments conducted in this work, adding such visualizations for all model performances proved challenging within the time constraints. We will, however, include the trading trajectories for agent models in a future version of the manuscript.
> 3.	We chose JPMorgan and Silver for testing due to their availability of a relatively large number of news articles included in the large-scale (1.85M) financial news dataset [1]. This ensured we had consistent and relevant data for analyzing their performance in the context of our study. Also, the large-scale nature of work required more resources for experimentation, so we were convinced to use only two mentioned stocks based on the number of available news articles in the given news dataset.
>
> [1] https://huggingface.co/datasets/ashraq/financial-news

---

> > ### Comment · Reviewer_MGL7 · 2024-12-02
> >
> > Thank you for your effort in addressing my questions. However, I would like to maintain the overall score because I still don't see significant novelty and advantage over the existing works on applying LLM for stock trading.

---

> > > ### Author Response · Authors · 2024-12-03
> > >
> > > Thank you for your valuable feedback.
> > > We greatly appreciate the time and effort you have taken to review our work and provide constructive comments. In this response, we aim to address your concerns thoroughly and highlight the key strengths and novel contributions of our approach. Specifically, we will:
> > >
> > > 1.	Provide a concise summary of the strengths of our proposed framework.
> > > 2.	Compare our work with related studies to clarify its unique advantages.
> > > 3.	Address specific concerns raised regarding the comparative aspects and demonstrate the applicability of our method in real-world scenarios.
> > > 4.	Offer our willingness to perform additional comparative analyses, including any specific studies you recommend, to further validate the distinct contributions of our work.
> > >
> > > We hope this structured response provides clarity and adequately demonstrates the novelty and practical significance of our research. Please feel free to share any additional feedback or suggestions for improvement.
> > >
> > > Strengths of Our Work
> > >
> > > 1.	Integration of RL and LLMs: Our study uniquely combines Deep Reinforcement Learning (DRL) with Large Language Models (LLMs) for trading strategies. This dual integration of textual and numerical data is rarely explored.
> > > 2.	Novel Algorithm – Stock-Evol-Instruct: We propose an innovative algorithm that generates adaptive instructions based on historical trends and real-time financial news, enhancing RL agents' decision-making capabilities.
> > > 3.	Diverse LLM Utilization: Our framework leverages six distinct LLMs (e.g., GPT-4o, LLaMA-3), showcasing adaptability across various market conditions.
> > > 4.	Fine-Tuned LLM Agents: Open-source models like Mistral-7B and LLaMA-3 are fine-tuned to act as autonomous, human-like trading agents.
> > > 5.	Enhanced News Analytics: By integrating financial news, our approach captures market sentiment and external factors, enabling context-aware trading.
> > >
> > > Advantages Over Compared Work
> > >
> > > 1.	Real-Time Decision Making: Unlike other works focused on regime adaptation (e.g., [1], [3]), our approach supports dynamic, daily trading decisions by integrating real-time news and historical data.
> > > 2.	Dynamic Instruction Generation: The Stock-Evol-Instruct algorithm introduces a novel, structured method for generating task-specific training datasets, absent in frameworks like FINCON [3].
> > > 3.	Broader Empirical Validation: We evaluate six LLMs on real-world datasets (e.g., Silver, JPMorgan), offering a more comprehensive analysis compared to works relying on simulations (e.g., [3]).
> > > 4.	Holistic Multi-Modal Analysis: Our integration of numerical and unstructured textual data provides a broader scope compared to works like FinVision [2], which emphasize specific data types.
> > > 5.	Focus on High-Frequency Trading: While other studies (e.g., [3]) target long-term portfolio strategies, our framework is explicitly designed for daily trading, addressing a critical gap in the literature.
> > >
> > > In Response to Your Concern
> > >
> > > Our contributions go beyond predictive modeling or static frameworks (e.g., [1], [2], [3]) by integrating LLMs with RL to dynamically optimize trading strategies. Key distinctions include:
> > >
> > > 1.	Real-Time Adaptability: Our framework enables RL agents to adjust strategies in response to evolving market conditions using live news data.
> > > 2.	Empirical Validation: Unlike [3], which relies on synthetic datasets, we validate our approach on real-world data, ensuring practical relevance.
> > > 3.	End-to-End Trading Agents: We provide fully autonomous agents capable of real-time decision-making and execution, surpassing the forecasting focus of FinVision [2].
> > >
> > > We are also happy to conduct further comparisons with other papers you suggest and demonstrate the advantages of our approach relative to any specific study you have in mind.
> > >
> > > Thank you again for your insightful comments, and we look forward to your feedback.
> > >
> > > [1]	M. Swamy, A. Shukla, and J. Purtilo, "Llm-based stock market trend prediction," 2023.
> > >
> > > [2]	 S. Fatemi and Y. Hu, "FinVision: A Multi-Agent Framework for Stock Market Prediction," in Proceedings of the 5th ACM International Conference on AI in Finance, 2024, pp. 582-590.
> > >
> > > [3]	Y. Yu et al., "FinCon: A Synthesized LLM Multi-Agent System with Conceptual Verbal Reinforcement for Enhanced Financial Decision Making," arXiv preprint arXiv:2407.06567, 2024.

---

### Official Review · Reviewer_19i3 · 2024-10-29

**Soundness:** 3
**Presentation:** 3
**Contribution:** 4
**Rating:** 8
**Confidence:** 5

**Summary:**

The paper explores a new approach to algorithmic trading by integrating LLMs with deep reinforcement learning to analyze and predict stock market trends.

It contributes meaningfully to algorithmic trading research by proposing an innovative model that combines LLMs and deep reinforcement learning. With refinements in prompt diversity, a deeper analysis of metrics, and further exploration of model limitations, the research could have a significant impact on the field of financial technology.

**Strengths:**

This paper presents a forward-thinking approach by merging LLMs and deep reinforcement learning, bridging the strengths of LLMs in handling unstructured data and deep RL's adaptability in a dynamic environment.

The authors outline their contributions well, particularly the development of Stock-Evol-Instruct.

Testing on real-world data (Silver and JPMorgan) is a strong point, as it validates the model's performance in practical scenarios.

The paper incorporates different prompt designs for LLM predictions, which offers a robust testing ground for prompt optimization.

**Weaknesses:**

There is an inconsistency in performance metrics (SR and ROI), which the authors highlight. The authors could delve deeper into explaining the contributions under which these inconsistencies occur and propose adjustments.

The study relies on a relatively small set of prompts for instruction generation, which may restrict the model's generalization across various market conditions.

The Stock-Evol-Instruct method includes in-depth evolving steps, which may introduce redundant complexity. Simplifying this process or providing a clearer rationale for each evolution step could improve understanding and replicability.

A more systematic analysis of how different LLMs align with specific trading objectives would strengthen the case for the chosen models.

**Questions:**

How do you plan to address the inconsistencies in the metrics in your evaluations?

What criteria were used for selecting the LLMs? How do they compare in performance beyond their general strengths in NLP?

Given the limited prompt variety, do you foresee exploring automated prompt generation or using techniques like semi-supervised learning to enhance prompt diversity?

---

> ### Author Response · Authors · 2024-11-27
>
> Dear Reviewer
>
> Thank you for your insightful comments and good discussion points.
>
> 1.	Inconsistency between SR and ROI: The inconsistency between the Sharpe Ratio (SR) and Return on Investment (ROI) could stem from several factors. While SR measures risk-adjusted returns, it may not fully capture the overall profitability of a trading strategy, especially in fluctuating market conditions where high risk can lead to large gains or losses. In contrast, ROI focuses purely on profitability, disregarding risk. This can result in discrepancies when models perform well in risk-adjusted terms (high SR) but fail to achieve consistent positive returns (low ROI), especially if they are not effectively capitalizing on market opportunities or if their risk exposure is misaligned with profitability. Additionally, the choice of model, prompt, and market characteristics could further contribute to these inconsistencies. Overall:
>
> a.	Some LLMs, such as GPT-4o or LLaMA variants, show high ROI values but may not consistently manage risk well, leading to lower Sharpe ratios. Similarly, some RL models like DDQN may have lower ROI but exhibit more stable performance, leading to higher Sharpe ratios.
>
> b.	The variations across different prompts (PT-1, PT-2, PT-3) may introduce changes in model behavior that affect ROI and SR differently. While some prompts improve ROI by capturing market trends, they may increase fluctuations, thereby reducing the SR. Overall, Prompt 3: Exemplar-based Forecasting, which uses examples to instruct the models to learn from examples, in most of the scenarios improves the SR but obtains very low ROI.
>
> 2.	For Selecting LLMs: Technically, we used the latest models that are capable of  Zero and few-shot prompting more effectively, we explored only a limited amount of LLMs and then for the study, we used the mentioned LLMs, due to their variants in the size and diversity in their training process.
>
> 3.	For stock-Evol-Instruct, we visualize the steps in Figure 2 (in Appendix E of the paper), clarifying the instruction generation. Where, the rationale behind for types of prompts in In-Depth evolving explained, in which 1) Adding Constraints has been utilized to ensure compliance with market regulations, 2) Depending incorporates dependencies between market factors for better context, 3) Concretizing refines abstract concepts into actionable signals, and finally 4) Increase Reasoning, enhances multi-step decision-making. All to build high-quality instructions for finetuning agents. We believe that the instruction generation method aimed to enrich the limited prompts, which in real-world scenarios can be challenging to make different prompts. However, we agree that semi-supervised learning techniques such as prompt-tunning [1] can be beneficial for generating more diverse initial prompts for stock-Evol-Instruct.
>
>
> [1] Lester, B., Al-Rfou, R., & Constant, N. (2021). The power of scale for parameter-efficient prompt tuning. arXiv preprint arXiv:2104.08691.

---

### Official Review · Reviewer_nQEt · 2024-11-04

**Soundness:** 3
**Presentation:** 3
**Contribution:** 3
**Rating:** 6
**Confidence:** 4

**Summary:**

This paper explores the integration of LLMs with RL to enhance algorithmic trading strategies in financial markets. By leveraging LLMs such as GPT-4o, LLaMA, and Mistral alongside DQN and DDQN models, the study demonstrates how LLM-driven insights from financial news can improve trading agents' decision-making.

It evaluates the effectiveness of various LLM-RL combinations on two case-study stocks, Silver (SLV) and JPMorgan (JPM), measuring performance through metrics like Return on Investment (ROI) and Sharpe Ratio (SR). Results indicate that LLMs integrated with RL strategies can outperform traditional RL alone by providing contextual market insights.

In the experiment, LLaMA-3 showed higher accuracy in trade prediction, while Mistral-7B excelled in profitability. It also  emphasizes the role of prompt design in optimizing model outputs and addresses the limitations of prompt diversity, suggesting future expansion.

**Strengths:**

1. Originality: The paper presents an innovative combination way of LLMs with RL algorithms and it expands the applicability of LLMs in quant trading. Compared to the previous work of applications in LLM and RL in financial trading, like FinMem, FinGPT and FinRL, which focuses on either financial LLMs—using NLP techniques to interpret market sentiment and to output trading decision—or DRL models designed for financial decision-making based purely on quantitative market data, this study bridges these two areas.

2. The study is thorough in its design and evaluation, offering clear descriptions of methodologies, metrics (ROI, Sharpe Ratio), and prompt types (zero-shot, instruction-based, and exemplar-based).

3. The paper is well-structured, with distinct sections dedicated to prompt design, trading environment, evaluation metrics, and empirical results.

4. The significance of this work lies in its contribution to enhancing algorithmic trading strategies through a novel combination of LLMs and DRL. Though still exploratory, this combination highlights a useful direction for future research into more nuanced decision-making systems in finance.

**Weaknesses:**

Absence of Baseline Comparisons: While the paper demonstrates that LLM-RL combinations outperform traditional RL models, it lacks comparisons to other established baselines or hybrid approaches in financial NLP or multi-modal models for trading, such as FinRL, FinMem.

**Questions:**

1. Would the authors consider adding other baselines in future work?

2. Given the importance of risk management in financial trading, could the authors elaborate on any plans to integrate risk-sensitive Reinforcement Learning (RL) approaches?

---

> ### Author Response · Authors · 2024-11-27
>
> Dear Reviewer,
>
> We appreciate your positive and insightful comments and suggestions for a very interesting integration with risk-sensitive RL.
>
> 1. Regarding the baseline models, we adapted FinRL [1], and FinGPT [2] models. FinRL is an open-source framework that establishes deep reinforcement learning strategies (all 5 models –  DDPG, TD3, A2C, SAC, and PPO – in the backend), where we obtained a better performance for JPM stock using PPO, and for SLV using TD3 models. FinGPT is an open-source financial LLM, aimed to be used in practical scenarios for a variety of goals. The experimental results indicated that (according to the table-4), domain-specific finetuning of FinGPT might not significantly contribute to the fully NLP-based finetuning agents, but can be useful if it adapts the instruction methodology introduced in this work, which requires further investigations.
>
> 2. The integration of the risk-sensitive RL approach. We agree that risk management is a critical aspect of financial trading, and integrating risk-sensitive approaches into our models is an important area for future exploration. One direction could involve incorporating risk-adjusted reward functions, such as Conditional Value-at-Risk (CVaR), into the RL training process. This would guide agents to make decisions to minimize exposure to risk, aligning the agent's strategy with real-world risk management practices. For this, using the LLM-as-a-judge [3] methodology could be beneficial. Where on one side we will have our agent as a model to be finetuned and on the other side more powerful LLM (such as the O1-Previous model, introduced by OpenAI) as a CVaR model. The prompt engineering with human evaluation can allow us to see how the evaluator model works, which later can be used to provide feedback for agents to explore the trading environment to take controlled risk trading. But this is an unexplored area for financial trading, which can take more effort as we don't know how closed-source LLMs can be suitable for such tasks.
>
>
> [1] Liu, X. Y., Yang, H., Chen, Q., Zhang, R., Yang, L., Xiao, B., & Wang, C. D. (2020). FinRL: A deep reinforcement learning library for automated stock trading in quantitative finance. arXiv preprint arXiv:2011.09607.
>
> [2] Yang, H., Liu, X. Y., & Wang, C. D. (2023). Fingpt: Open-source financial large language models. arXiv preprint arXiv:2306.06031.
>
> [3] Zheng, L., Chiang, W. L., Sheng, Y., Zhuang, S., Wu, Z., Zhuang, Y., ... & Stoica, I. (2023). Judging llm-as-a-judge with mt-bench and chatbot arena. Advances in Neural Information Processing Systems, 36, 46595-46623.

---

### Official Review · Reviewer_Dgbh · 2024-11-05

**Soundness:** 2
**Presentation:** 1
**Contribution:** 1
**Rating:** 1
**Confidence:** 5

**Summary:**

This paper presents a novel approach to algorithmic trading that combines Deep Reinforcement Learning (DRL) with Large Language Models (LLMs). The author introduce "Stock-Evol-Instruct," a new instruction generation algorithm that helps optimize trading strategies by incorporating LLM-driven insights from financial news. The study examines six different LLMs, including LLaMA-2, LLaMA-3, Mistral-7B, Falcon-7B, OpenELM, and GPT-4o, integrating them with DQN and DDQN reinforcement learning methods. Testing their approach on Silver (SLV) and JPMorgan (JPM) stocks, the paper found that LLM-enhanced trading strategies often outperformed traditional RL methods alone, with some models achieving significant improvements in Sharpe Ratio and ROI. The fine-tuned Mistral-7B model, in particular, showed strong performance with ROIs of 53.15% and 48.36% for JPM and SLV respectively. The paper demonstrates the potential of combining LLMs with reinforcement learning to create more sophisticated and effective algorithmic trading systems.

**Strengths:**

The paper studies an important problem, i.e., automatic financial trading with LLM-enhanced reinforcement learning.
The author conducted extensive experiments to validate the proposed approach.

**Weaknesses:**

1. The literature review is not thorough, many relevant papers in LLM/AI for finance and trading area are missing. Some paper I found relevant are listed below, but I believe there are more to be included.

2. The writing of the paper can be improved. However, the main problem is the novelty of the work is really limited due to that the author did not conduct a thorough literature review for the field of RL/DL/LLM for trading. There are so many important and related work in this field that should at least be included as baselines for comparison. I recommend authors first conduct a comprehensive literature review in this field, then conduct comprehensive and fair baseline comparisons.


Yu Y, Li H, Chen Z, et al. FinMem: A performance-enhanced LLM trading agent with layered memory and character design[C]//Proceedings of the AAAI Symposium Series. 2024, 3(1): 595-597.

Yu Y, Yao Z, Li H, et al. FinCon: A Synthesized LLM Multi-Agent System with Conceptual Verbal Reinforcement for Enhanced Financial Decision Making[J]. arXiv preprint arXiv:2407.06567, 2024.

Yuan Z, Liu J, Zhou H, et al. LEVER: Online Adaptive Sequence Learning Framework for High-Frequency Trading[J]. IEEE Transactions on Knowledge and Data Engineering, 2023.

Yuan Z, Liu H, Hu R, et al. Self-supervised prototype representation learning for event-based corporate profiling[C]//Proceedings of the AAAI Conference on Artificial Intelligence. 2021, 35(5): 4644-4652.

**Questions:**

Please refer to my questions above.

---

> ### Author Response · Authors · 2024-11-27
>
> Dear Reviewer,
>
> Thanks for sparing the time and providing valuable feedback for improving the manuscript.
>
> 1. We conducted a deeper literature review over 12 relevant papers as listed below (including the 4 papers that you suggested). The highlighted blue text has been added to the related work section of the paper.
>
> 2. After a careful review of the literature we, extended the experimentations for the FinGPT[10] and FinRL [5] models.  These models played a baseline model role for comparison of proposed methods. The results have been added to Table 3 (for comparison of Q-learning and LLMs models) and Table 4 (for comparison of trading agents).  FinGPT [10] as an open-source LLM, takes a data-centric approach with the aim of providing researchers and practitioners with accessible and transparent resources to develop financial LLM. As a potential LLM for the financial sector, we used this model as a baseline for both Q-Learning and agen models. Moreover, 5 RL models introduced by FinRL[5] have been explored for agent model comparison. FinRL uses deep reinforcement learning models with DDPG, TD3, A2C, SAC, and PPO models as a backbone. We explored all the models per stock data in this study and we reported the best model per stock data for comparison of agent models. Overall 28 more experiments were conducted to conclude the baseline setups. We will continue exploring more baselines in a timely manner for more empirical setups
>
> As baseline models provided additional context for evaluating the fine-tuned agents (results presented in Table 4). FinRL, a fully RL-based model, yielded a minimal ROI of 0.04% for JPM and 7.33% for SLV, indicating limited profitability when relying solely on stock time-series data. FinGPT, a fine-tuned model on trading datasets, exhibited a negative ROI of -8.28% for JPM and -20.58% for SLV, demonstrating challenges in leveraging natural language data for consistent profitability. However, for FinRL (the results presented in Table 3), generally lower, performance compared to the LLM-enhanced strategies across most evaluation metrics observed. While they exhibited reasonable SR in some cases, such as for JPMorgan (JPM), they often struggled with achieving consistent positive ROI.
>
> [1] Yu, Y., Li, H., Chen, Z., Jiang, Y., Li, Y., Zhang, D., ... & Khashanah, K. (2024, May). FinMem: A performance-enhanced LLM trading agent with layered memory and character design. In Proceedings of the AAAI Symposium Series (Vol. 3, No. 1, pp. 595-597).
>
> [2] Yu, Y., Yao, Z., Li, H., Deng, Z., Cao, Y., Chen, Z., ... & Xie, Q. (2024). FinCon: A Synthesized LLM Multi-Agent System with Conceptual Verbal Reinforcement for Enhanced Financial Decision Making. arXiv preprint arXiv:2407.06567.
>
> [3] Yuan, Z., Liu, J., Zhou, H., Zhang, D., Liu, H., Zhu, N., & Xiong, H. (2023). LEVER: Online Adaptive Sequence Learning Framework for High-Frequency Trading. IEEE Transactions on Knowledge and Data Engineering.
>
> [4] Yuan, Z., Liu, H., Hu, R., Zhang, D., & Xiong, H. (2021, May). Self-supervised prototype representation learning for event-based corporate profiling. In Proceedings of the AAAI Conference on Artificial Intelligence (Vol. 35, No. 5, pp. 4644-4652).
>
> [5] Liu, X. Y., Yang, H., Chen, Q., Zhang, R., Yang, L., Xiao, B., & Wang, C. D. (2020). FinRL: A deep reinforcement learning library for automated stock trading in quantitative finance. arXiv preprint arXiv:2011.09607.
>
> [6] Wang, M., Izumi, K., & Sakaji, H. (2024). LLMFactor: Extracting Profitable Factors through Prompts for Explainable Stock Movement Prediction. arXiv preprint arXiv:2406.10811.
>
> [7] Fatouros, G., Metaxas, K., Soldatos, J., & Kyriazis, D. (2024). Can large language models beat wall street? unveiling the potential of ai in stock selection. arXiv preprint arXiv:2401.03737.
>
> [8] Zhang, W., Zhao, L., Xia, H., Sun, S., Sun, J., Qin, M., ... & An, B. (2024). FinAgent: A Multimodal Foundation Agent for Financial Trading: Tool-Augmented, Diversified, and Generalist. arXiv preprint arXiv:2402.18485.
>
> [9] Li, Y., Yu, Y., Li, H., Chen, Z., & Khashanah, K. (2023). TradingGPT: Multi-agent system with layered memory and distinct characters for enhanced financial trading performance. arXiv preprint arXiv:2309.03736.
>
> [10] Yang, H., Liu, X. Y., & Wang, C. D. (2023). Fingpt: Open-source financial large language models. arXiv preprint arXiv:2306.06031.
>
> [11] Yuan, H., Wang, S., & Guo, J. (2024). Alpha-GPT 2.0: Human-in-the-Loop AI for Quantitative Investment. arXiv preprint arXiv:2402.09746.
>
> [12] Wang, S., Yuan, H., Ni, L. M., & Guo, J. (2024). QuantAgent: Seeking Holy Grail in Trading by Self-Improving Large Language Model. arXiv preprint arXiv:2402.03755.

---

### Meta-Review · Area_Chair_o8eF · 2024-12-20

**Metareview:**

The paper introduces a novel approach to algorithmic trading by combining Large Language Models (LLMs) with Reinforcement Learning (RL), specifically a Double Deep Q-Network (DDQN). The authors aim to enhance RL trading strategies with LLM-generated rewards derived from financial news. While the integration of LLMs with RL is an interesting contribution, several reviewers raised concerns about the novelty of the work, noting that similar approaches have been explored in prior research, such as FinMem, FinGPT, and FinRL. The paper does not sufficiently differentiate itself from these studies, and this lack of novelty was a significant factor in maintaining the reviewers' original scores.

The experimental design is strong, with real-world data and a broad evaluation of different LLMs and prompts. However, the paper lacks baseline comparisons with traditional strategies like "buy and hold" or models that use only LLMs for stock trading. Additionally, the relationship between the LLM and RL components is not explained clearly enough, and more effort is needed to illustrate how their combination improves trading performance. The choice of stocks (Silver and JPMorgan) is not well-justified, and the paper would benefit from additional visualizations, particularly of trading trajectories, to make the results more interpretable.

While the paper’s methodology is innovative, the reviewers expressed that the contribution would be stronger with clearer explanations, baseline comparisons, and a stronger justification of the stock selection. These revisions are necessary to fully demonstrate the novelty and impact of the proposed approach.

**Additional Comments On Reviewer Discussion:**

After the rebuttals, reviewers acknowledged the authors' efforts but maintained their original assessments. They appreciated the novelty of combining LLMs with RL for trading but still felt the paper did not offer significant advantages over existing work in the field. Concerns were raised about the lack of clear novelty, insufficient baseline comparisons, and unclear integration between LLMs and RL. Some reviewers suggested including additional baselines, exploring risk-sensitive RL approaches, and clarifying the choice of stocks used in experiments. Despite the authors' clarifications, reviewers remained unconvinced about the paper's contribution and its differentiation from prior studies.

---

### Decision · Program_Chairs · 2025-01-22

Reject